# Risk of Lameness in Dairy Cows with Paratuberculosis Infection

**DOI:** 10.3390/ani9060339

**Published:** 2019-06-10

**Authors:** Joshua Smith, Steven van Winden

**Affiliations:** Pathobiology and Population Sciences, Royal Veterinary College, Hawkshead Lane, Hatfield AL9 7TA, Hertfordshire, UK; josmith2@rvc.ac.uk

**Keywords:** paratuberculosis, dairy cows, lameness

## Abstract

**Simple Summary:**

Johne’s disease (JD) is an important disease affecting cows in the UK dairy industry, as is lameness, and both cause milk production losses and cows losing weight. The aim of this work was to see if cows with JD were more likely to be lame and if so, what the order of occurrence of these two events is. We looked at the JD and lameness records of two farms and paired 98 JD cows (half of them with a high response to the test for JD) and compared them with herd mates. We evaluated the timing and the proportion of lameness in JD-positive cows compared to their controls and proportion of lameness before and after the first JD-positive test. JD cows turn lame on average three months earlier and are lame 2.7 times more often than non-JD cows. Further, high positive cows were 2.8 times more likely to develop lameness after JD diagnosis compared to medium positive cows. Results of this study suggest that there is a link between JD and lameness and that JD precedes lameness. The underlying mechanisms for this association remain unknown and were not the scope of this study.

**Abstract:**

Johne’s disease (JD) is an important disease affecting the UK dairy industry, as is cattle lameness. An association between JD and lameness has been suggested; however, little evidence exists to support this. The purpose of this study was to determine if cows affected by JD were more likely to be lame and if so, what the temporal association is. Retrospective dairy cow mobility and JD status (based on milk ELISA) data were obtained from two farms of 98 JD cows (49 high and 49 medium positive) and their matched controls. We evaluated the timing and the proportion of (chronic) lameness in JD-positive cows versus controls and proportion of lameness before and after the first ELISA positive test. Compared to their controls, JD cows are lame more often (Odds Ratio = 2.7 (95% Confidence Interval = 1.2–6.0) *p* = 0.017) and became lame on average three months earlier (*p* = 0.010). High positive cows were more likely to develop lameness after seroconversion (OR = 2.8 (95% CI = 1.1–7.5), *p* = 0.038) versus medium positive cows. Results of this study suggest that there is a link between JD and lameness and that JD precedes lameness. The underlying mechanisms for this association remain unknown and were not the scope of this study.

## 1. Introduction

Johne’s disease (JD) is caused by the gram positive, acid fast, intracellular organism *Mycobacterium avium* subspecies *paratuberculosis* (*MAP*). Typically, cows get infected as young calves, with rates of infection decreasing significantly with age and the main risk period for infection being the first 6 months of life [1,2]. Although most ruminants can be affected, it is of particular importance to dairy cows. Despite an early-life infection, cows do not show clinical signs or subclinical losses until at least 2 years old; it is a chronic, progressive, and ultimately fatal granulomatous enteritis [1,3,4]. 

*MAP* is endemic to the dairy herd in the United Kingdom (UK); a survey with 225 randomly selected dairy farms, stratified by region and herd size, identified a true herd prevalence of 68% (95% CI: 61–76) in the national dairy herd [5]. No recent prevalence data are available of Johne’s at individual cow level for the UK, but a study in abattoirs in the Southwest of England found 3.5% of culled cows had evidence of subclinical JD [6]. Clinically, the disease causes chronic diarrhoea, weight loss, and oedema, with clinical signs tending to manifest in the late stages of the disease [7]. 

Very few cows develop clinical signs, with estimates of only 3 in 10,000 showing clinical signs every year in England and Wales [8]. Clinical and subclinical effects of JD impact directly and indirectly through economic losses on dairy farms. In the UK, it was estimated that costs to a hypothetical 200-cow herd was in the region of £7000–£11,000 in losses forgone [4]. Losses are due to premature culling, reduced milk production, and decreased slaughter value [9]. 

Upon initial oral exposure of the calf, *MAP* is incorporated into the mucosal lymphoid tissue, the majority of which is located in the ileum. This explains the main site of lesions [7]. Here, *MAP* is taken up by macrophages and a cell-mediated immune response is initiated, with the bacteria being contained within microscopic granulomas [10,11]. Infected cows will remain in a subclinical stage of infection for several years, with diffuse granulomatous lesions developing [7]. There appears to be progression from cell-mediated immune response to a humoral mediated immune response with the production of antibodies [12]. The humoral-based immune response is not protective against *MAP* and precedes the excretion of *MAP* pathogens in the faeces. The presence of antibodies to MAP is the evidence that the shift in immune response has occurred. The reasons or triggers for this change and, hence, progression of MAP infection into JD is currently unclear, but underlying genetic factors and exogenous stress factors such as parturition, malnutrition, or other diseases (e.g., mastitis, lameness) have been theorised [13]. 

Diagnosis of JD can be performed using different methods. Common techniques include faecal culture or polymerase chain reaction (PCR) and serum or milk antibody enzyme-linked immunosorbent assay (ELISA). Although all of these tests have a specificity of above 90%, they are limited by low sensitivity [14]. The sensitivity of a single milk ELISA test is relatively low for milking heifers, around 40%, but increases with age to 77.5% (95% CI: 71.0–82.9). In contrast to single tests, repeated milk ELISA testing allows risk profiling of the dairy cows and has therefore become current practice to pick up cows earlier, before becoming clinical, which allows improved calving management with the aim of reducing transfer of *MAP* infections to the calves [15]. 

Whilst it is important to focus on JD and the direct consequences, as there is a substantial impact on the UK dairy industry, links to other disease processes have been investigated. It has been shown that cows seropositive for *MAP* are associated with a higher individual cow milk somatic cell count and are believed to be more prone to mastitis [16,17]. There are also indications that JD decreases fertility, with ELISA positive cows having on average 28 extra days open compared to seronegative herd mates [18]. Reasons for the relationship are unknown at present; however, proposals include negative energy balance (NEB) and reduced cellular immunity seen with Johne’s disease [13,16].

Anecdotally, it has been remarked that cows with JD are more likely to be lame than those not affected. However, there is very limited scientific evidence to support this observation. Raizman et al. [19] found that the most common clinical sign seen in cows with JD was lameness. Villarino and Jordan [20] found that Johne’s disease-affected cows were five times more likely to be lame than non-affected cows; however, the temporal association remains unclear.

Lameness is an important condition affecting cows. In terms of cost to the dairy industry, it is second only to mastitis [21]. There are a wide range of causes of lameness, which can be divided into infectious (e.g., digital dermatitis and foul) and non-infectious (e.g., sole ulcers or white line disease) causes [22]. Current estimates in 2010 place the prevalence of lameness in the UK at roughly 36%; this is an increase from 20.6% in 1990 [23,24]. It is due to the economic and welfare consequences of lameness and increasing prevalence that there is an industry-wide desire and drive to reduce the overall impact of lameness. Mobility scoring is a useful tool in assessing the status of cows in the herd [25]. The results of the scoring can be used to formulate action plans, generate treatment lists and monitor herd level lameness. 

As both Johne’s disease and lameness have a major impact on the UK dairy industry, it is therefore important to understand if there is a (temporal) relationship between the two and what this relationship signifies. This will further assist in improving animal welfare and minimising losses. The aim of the study was to investigate potential links between Johne’s disease and lameness. This was done by identifying whether cows with JD (milk ELISA positive) are more likely to be lame than matched control (ELISA negative) cows and by determining the temporal sequence of events. 

## 2. Materials and Methods

### 2.1. Cow Selection

Two farms in Southern England were selected for the study and consisted mainly of Holstein Friesian cows which were milked twice daily. Herd A had 130 milking cows and herd B had 370 milking cows. The farms are considered a convenience sample, and selection was based on the presence of Johne’s disease on the farm, routine lameness recording and having quarterly milk ELISAs for MAP tests done for at least one year. Mobility scoring on exit from the milking parlour was assessed on both farms by the Royal Veterinary College using the Agriculture and Horticulture Development Board (AHDB) Dairy 0–3 system [26]. In both herds, clinically lame cows were foot-trimmed therapeutically on a monthly basis as well, and all cows were routinely foot-trimmed at dry-off.

Herd A block-calved in the autumn; all cows had pasture access during the summer with maize silage buffer feed before daily turn-out. Winter ration consisted of 50%–50% maize- and grass silage offered as a mixed ration. The cows were fed concentrates in the milking parlour. Average 305-day milk yield was 7652 kg, with 4.21% fat, 3.34% protein and 148,000 cells/mL somatic cell count. Throughout the year, copper sulphate footbaths were used three times a week, and mobility scoring took place every month; clinical lameness was at an annual incidence rate of 21%. 

Herd B calved year-round; low yielding cows had access to pasture in summer, whilst the high yielders were in receiving 60%–40% maize- and grass silage offered as a mixed ration. The cows were fed concentrates in the milking parlour. Average 305-day milk yield was 8710 kg, with 4.04% fat, 3.23% protein and 224,000 cells/mL somatic cell count. Formalin footbaths were used every two days year-round, and lameness scoring was performed every two months; clinical lameness was at an annual incidence rate of 13%.

Johne’s testing was performed on both farms by individual milk antibody ELISA through the NMR HerdWise scheme. The test used was a commercial milk ELISA, produced by IDEXX; Pourquier *Mycobacterium paratuberculosis* Screening Antibody Test [27] (IDEXX Laboratories Inc., Westbrook, ME). The results were read following the manufacturer’s guidelines, using a cut-off of 20%: Tests with a sample-to-positive ratio (S/P) of 20% or above were considered positive, high positive cows has an S/P ratio of 30% or above, and negative cows never had an ELISA result over the S/P 20% threshold. The calendar date of the first ELISA positive result was established for all Johne’s positive cows and was used to identify a matched control cow. The pairing within the herd was first done by parity, and when multiple controls were available, the cow with the closest calving date was selected.

For each ELISA positive cow and its paired control, individual mobility scores were assessed, using the AHDB Dairy 0–3 system; cows with a score 2 or 3 were considered as lame. Cows were considered chronically lame if they had two or more subsequent lame recordings on mobility scores. Based on the date of turning ELISA positive, it was determined whether cows were lame before or after this event.

### 2.2. Statistical Analysis

Data were entered into Microsoft Excel^®^ for data analysis with statistical analysis performed in GraphPad Prism^®^ ((GraphPad Software, San Diego, California, USA)) and IBM SPSS Statistics^®^ (IBM Corporation, Armonk, New York, USA). The statistical significance cut-off was set at *p* < 0.05. The median parity was 3 at the point of seroconversion, and 22 months since first calving. The cows were 219 days in milk (95% confidence interval: 173–266) at point of seroconversion.

A Kaplan Meier survival analysis was performed for all cows in the study, comparing ELISA positive (S/P ratio >20%) cows with their matched controls. Date of entry was set at either date of first calving or the beginning of mobility scorings on the farm, whichever was later. The hazard event was set as the first lame score. The number of months was calculated to the hazard event for every cow. Cows that did not have any event of lameness were right censored. GraphPad Prism^®^ was used to calculate survival times to first lameness and log rank hazard ratio of the ELISA positive group compared to the control group. Log rank statistical testing was applied to compare the two curves.

We assessed whether the occurrence of lameness differed between Johne’s ELISA positive cows and their controls; conditional logistic regression was performed to account for herd and parity matching. In addition, to evaluate whether the lameness developed before or after the Johne’s seroconversion, we ran logistic regression comparing the lameness rates pre- and post-seroconversion between the medium and high positive cows. Farm and lactation number were forced into this model to account for these factors. As a measure of magnitude of association, odds ratios (OR) and 95% confidence intervals (CI) were calculated [28].

### 2.3. Ethical Approval

Informed consent was obtained from each farmer contributing to the study. The Clinical Research and Ethical Review Board (CRERB) of the Royal Veterinary College, University of London has examined and approved the research protocol (2016-U15). 

## 3. Results

Overall, 98 ELISA positive cows were present on both farms. Of the ELISA positive cows, 49 had a high positive (S/P > 30%) result and 49 were in the medium positive group (S/P 20–30%). The median parity was 3 at the point of seroconversion, and 22 months since first calving. In total, 103 cows were classed as lame during the study (Table 1), and 30 were classed as chronically lame. 

The Kaplan Meier survival analysis indicated that median survival time (i.e., time from entry into the study until the first lameness event) for the ELISA positive (S/P > 20%) and control group was 22 and 25 months, respectively. The survival curves are shown in Figure 1. Log rank statistical testing applied to these survival curves revealed that there was a significant difference between the ELISA positive cows and their controls, *p* = 0.010 (Figure 1). The log rank hazard ratio for the ELISA positive group compared to control was 1.7 (95% CI = 1.1–2.6). 

The conditional logistic regression showed that seropositive cows were more likely to be lame (OR = 2.7 (95% CI = 1.2–6.0) *p* = 0.017). Looking specifically at the cows with a high serological response (S/R ratio >30%), an even higher proportion of lameness was found compared to their matched controls (OR = 3.6 (95% CI = 1.1–11.2), *p* = 0.029). The medium positive cows were, however, not different to their controls regarding their lameness incidence (OR = 2.0 (95% CI = 0.6–6.2) *p* = 0.250).

A slightly higher number of cows that were ELISA positive had a chronic episode of lameness than the control group, 17.3% (n = 17) compared to 13.3% (n = 13). Based on the conditional logistic regression, the difference was not statistically significant, *p* = 0.160 with OR = 2.3 (95% CI = 0.7–7.0).

Of the seropositive cows, 35 cows turned lame after their first ELISA positive test, compared to 22 cows being lame before their first ELISA positive test. Comparing high positive cows to medium positive, whilst controlling for farm and lactation number, there was not difference in the lameness prevalence before seroconversion between the groups (*p* = 0.633). High positive cows were, however, more likely to develop lameness after seroconversion (OR = 2.8 (95% CI = 1.1–7.5), *p* = 0.038), compared to medium positive cows.

## 4. Discussion

It was observed that ELISA positive cows had a higher proportion of lameness and were 2.7 (S/R > 20%) or 3.6 (high seropositive) times more likely to be lame than the control group. These findings have also been reported by Villarino and Jordan, who reported a slightly lower risk ratio (1.67). In addition, on average, Johne’s positive cows go lame three months earlier than their matched controls and at a similar time of seroconversion, both 22 months into their productive lives. The reason for an increased level of lameness in Johne’s affected cows could be three-fold; through decreased horn quality, increased NEB or reduced immune function. As NEB occurs at the start of the lactation, and both the lameness and JD seroconversion in this study occurred toward the end, the NEB could be a confounding factor that predisposes cows to developing both JD and lameness, through the impact of an impaired immune response in cows with NEB. 

Looking at the sequence of events, we found that the high JD-positive cows were more likely to be lame after their first positive ELISA rather than before. Therefore, we can assume that Johne’s disease predisposes cows to developing lameness. We do need to consider confounding factors, such as NEB, which may predispose cows for both seroconversion and lameness development. The main confounding factors for both lameness and JD seroconversion we have accounted for by the matching on lactation number and days in milk of the case and control herd mates, leaving other factors, such as milk yield and associated NEB, that could explain both events [29,30,31].

It is thought that cows with Johne’s disease are more prone to periods of severe NEB. It is still uncertain if subclinical lesions are sufficient to cause reduced absorptive capacity and thus cause NEB, but it is presumed that mild reductions in function are enough to increase the risk of severe NEB [13,18]. Seroconversion is associated with clinical progression, which is characterised by weight loss [32], and cows with lower body condition scores (BCS) are predisposed to lameness [33,34]. During periods of NEB, fat is mobilised from stores around the body. The digital cushion in cows consists partially of fat and is thought to be mobilised during NEB with the protective effects of the digital cushion on the hoof being reduced [29]. In cows with Johne’s disease undergoing NEB, the digital cushion depletes more quickly of fat and consequently increases the chance on lameness.

Johne’s disease has been thought to affect keratin quality in sheep (wool and horn) [35,36]; however, evidence of this link is limited and conflicting [37,38]. In keratin, biotin has been shown to determine the quality of the hooves [39,40,41], and biotin is absorbed in the small intestine, the primary lesion site for JD. In humans, biotin supplementation is required in patients with short gut syndrome or other causes of malabsorption [42]. Conceivably, cows with JD are not as efficient at absorbing biotin from their diet and consequently have poorer quality hooves. 

It has also been suggested that cows with JD have a reduced cellular immunity as a potential explanation for increased mastitis and cell counts [16]. A reduced cellular immunity could make seropositive cows more susceptible to infectious causes of lameness. Sadly, the cause of the lameness was not recorded routinely on the farms that were studied, so the role of this potential explanation cannot be fully explored. Neither did we measure the NEB, biotin levels or the cellular immunity of our cows, as it was not the initial scope of this retrospective study. The current study suggests that there is reason to include these in further studies, or indeed evaluate whether biotin supplementation reduces the lameness in JD-positive cows more than in the average, JD-negative cow.

Johne’s disease and lameness have huge financial and welfare impacts on farms. The current suggested temporal link suggests that the cost of JD may need to include the cost of lameness in the dairy herd, at least partly. The economic burden to the UK dairy industry of lameness is rated at £323 per case, and second only to mastitis [21,43] In addition, specific targeted interventions focusing on reducing the impact of JD can be supported by the potential reduction of lameness cases and to minimise losses and suffering.

## 5. Conclusions

The results from this investigation show that, controlled for farm and parity, Johne’s disease antibody ELISA positive cows turn lame significantly earlier and are more likely to be lame than JD-negative cows. High positive cows also have more lameness after the first ELISA positive result than cows that test medium positive. The current study did not show a causal relationship between JD and lameness, but the (temporal) association between the two conditions may help us to speculate on the development of each. Reasons for the association could be Johne’s disease predisposing to lameness or both conditions having a common underlying predisposing factor, such as a negative energy balance earlier in the lactation. Knowing the relationship between the two conditions allows us to consider specific on-farm interventions to minimise losses and reduce negative impacts of Johne’s disease on compromised welfare.

## Figures and Tables

**Figure 1 animals-09-00339-f001:**
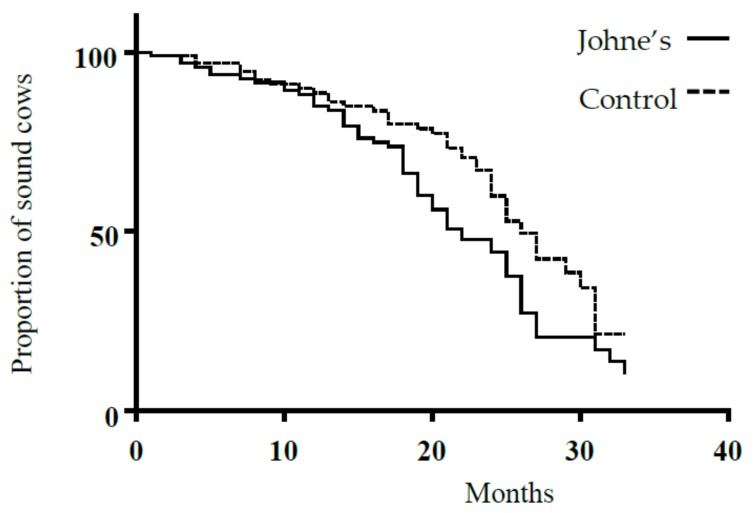
Time elapsed until the occurrence of a lameness event in cows diagnosed with Johne’s disease compared to their lactation matched seronegative controls.

**Table 1 animals-09-00339-t001:** Number of lame and sound cows based on Johne’s status.

Lameness Status	Negative Cows (n = 98) (<20% S/P)	Medium Positive Cows (n = 49) (20% < S/P < 30%)	High Positive Cows (n = 49) (>30% S/P)
Lame cows (of which chronic)	46 (13)	27 (10)	30 (7)
Before/after Johne’s positive	NA	13/14	9/21
Sound cows	52	22	19

NA: not applicable.

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
