# Peer review of "Risk of Lameness in Dairy Cows with Paratuberculosis Infection"

_animals, 2019, doi:10.3390/ani9060339_

Round 1

Reviewer 1 Report

General comments:

Did you check for presence of other diseases in cows with JD? For example, mastitis, infertility, displaced abomasum, retained placenta, ketosis, milk fever.

How come the Simple Summary and the Abstract are the same length.

Introduction should be improved. The authors need to focus their introduction on the potential links between JD and lameness not describing what JD is. Everything that has been published with regards to JD and lameness needs to be presented in the Introduction section.

Given that the data reported are very small (A table and a graph), I would suggest this manuscript to be converted into a short communication.

Specific comments:

Line 14: This simple summary is for lame people. Therefore, they have no idea what is JD high and medium positive. So, explain or remove.

Line 18: Check the sentence. It seems grammatically wrong.

Line 19-20: “after seroconversion” again lame people would not understand what this is.

Line 34: What was the OR of medium positive cows.

Line 42: Be consistent, use JD instead of MAP. You already indicated what JD is, so continue with the same terminology.

Line 48: Fix the sentence. There are errors… “to the in the”.

Line 88: What do you mean by the sequence of events? You are not dealing with the sequence of events in your study.

Lines 109, 116: Indicate what was milk production and BCS for cows involved in the study. Also, indicate what was the diet used for those two farms. Readers also would like to know what was the DIM and the average parity of the cows in the study. Otherwise this would be a very superficial study, and needs to be converted into a short communication.

Line 158: Information about parity should have been given to the Materials and Methods section.

Line 177: Info about the lactation number of the cows in the study should have been given first at the Materials and Methods.

Line 187: There more than three reason for lameness. Some other reasons for lameness (unrelated to JD) include: 1) endotoxin translocation, 2) histamine, and 3) high grain diets, which have no direct relationship with JD. JD might be another reason but not the main reason for lameness. To state that JD and lameness have a similar causology is totally wrong. Lameness affects up to 36% of the herds, which is not the case for JD. How do the authors explain that? So, we have to be very careful and not to state a common etiology for bothb diseases.

Line 188: Sentence is missing a verb.

Line 190: What might be that common confounding factor for JD and lameness? Elaborate what are your thoughts. Even a mere speculation.

Line 194: What do you mean by “…predispose cows for both seroconversion and …”? Seroconversion is not a disease. You should indicate JD, instead.

Conclusions: The authors need to indicate that “currently there is no known causal relationship between JD and lameness and that all their discussions are simply speculations”. We should be very careful with regards to stating a common etiology of JD and lameness. We know that JD is an infectious disease. Lameness is not a disease in itself; it’s just a sign of an inflammation of the hoof. It might be caused also even by mechanical injury (environmental factors) or by feeding high amounts of concentrate (grain).

Author Response

Thank you for your thorough review, in the annotation version of the resubmission you find your comments and suggestions addressed. Attached also our specific responses to the queries raised.

Reviewer 2 Report

The present paper investigates the potential links between Johne’s disease and lameness to understand what this relationship signifies.

The objectives of the study are of interest and fit well within the scope of the journal and, in overall, the research was performed with an adequate description of methodology used.

The introduction provides a good, generalized background of the topic that quickly gives the reader an appreciation of the disease’s importance and of the direct consequences on the UK dairy industry.

The work appears adequately conducted and the results reported are pertinent.

In my opinion, the manuscript could be accepted for publication.

Author Response

Thank you for your kind review, in the annotation version of the resubmission you find our responses to you and your peer-reviewers' comments.

Reviewer 3 Report

Respected Author,

I was reading your submission with interest. I have to agree that not many signals that could correlate with the presence of the lameness in dairy cows are studied sufficiently.

Paratuberculosis is one of the issues. As I went through the paper following notes comed to me:

Introduction

L.48 First part of the sentence - reformulate, not clear. (even not being native speaker)

L.102 Is the Aim based on the anecdotally hypothesis? (L.84)

M&M

L.116 Isn’t the Formalin officially forbidden as hazardous and potentially carcinogenous agent?

Results

Information about production level of Herd A and B is missing.

Figure 1. Low resolution - it could be improved.

Discussion

More information about economical impact of lameness on dairy industry is missing.

Lameness treatment economic weights were calculated recently for dairy cattle (not only Holstein:)

ICAR established WG on dealing with recording of lameness as part of the selection process.

Author Response

Thank you for your review, in the annotation version of the resubmission you find your comments and suggestions addressed as well a our response to other reviewers. Attached also our specific responses to the queries raised.

Round 2

Reviewer 1 Report

No further comments.